# Prevalence and risk factors of gross neurologic deficits in children after severe malaria: A systematic review and meta-analysis

Allen Eva Okullo [1,2,3]*, Simple Ouma [2,4], Chandy C. John [5],
Michael Boele van Hensbroek [1], Kevin Ouma Ojiambo [3,6], Caroline Otike [3,7],
Alison Annet Kinengyere [3,8], Andrea L. Conroy [2,5], Moses Ocan [3,9], Ekwaro A. Obuku [3,10],
Richard Idro [4]

1 Department of Paediatrics, Amsterdam University Medical Centers, University of Amsterdam, Amsterdam, The Netherlands, 2 Global Health Uganda, Kampala, Uganda, 3 Africa Center for Systematic Reviews & Knowledge Translation, College of Health Sciences, Makerere University, Kampala, Uganda, 4 Department of Paediatrics and Child Health, School of Medicine, College of Health Sciences, Makerere University, Kampala, Uganda, 5 Ryan White Center for Pediatric Infectious Diseases and Global Health, Indiana University School of Medicine, Indianapolis, Indiana, United States of America, 6 Clinical Epidemiology Unit, School of Medicine, College of Health Sciences, Makerere University, Kampala, Uganda, 7 Joint Clinical Research Centre, Kampala, Uganda, 8 Sir Albert Cook Library, College of Health Sciences, Makerere University, Kampala, Uganda, 9 Department of Pharmacology & Therapeutics, School of Biomedical Sciences, College of Health Sciences, Makerere University, Kampala, Uganda, 10 Faculty of Epidemiology & Population Health, London School of Hygiene & Tropical Medicine, London, United Kingdom

* a.e.okullo@amsterdamumc.nl

## Abstract

### Background

Children with severe malaria may develop gross neurologic deficit(s). We conducted a systematic review on the prevalence and risk factors of gross neurologic deficits after childhood severe malaria.

### Methods

The systematic review was conducted following PRISMA guidelines. Article search was conducted in MEDLINE, EMBASE, Web of Science, and Global Index Medicus. Studies included reported on prevalence and/or risk factors of gross neurologic deficits after severe malaria in children. Risk of bias analysis and heterogeneity assessment were performed using ROBINS tool and $I^2$-statistic, respectively. Data analysis was done using quantitative synthesis in R *ver*4.5.0 software, and narrative synthesis.

### Results

41 studies from 16 countries in Sub-Saharan Africa and Asia comprising 11,635 children were included in the analysis. Gross neurologic deficits included motor,

**Data availability statement:** All relevant data are within the paper and its Supporting information files.

**Funding:** This systematic review and meta-analysis has been funded by grants from the Fogarty International Center (https://www.fic.nih.gov), grant number D43 TW010928, received by CCJ and the National Institute of Neurological Disorders and Stroke (https://www.ninds.nih.gov), grant number R01 NS055349 received by CCJ. The funders did not play any role in the study design, data collection and analysis, decision to publish, or preparation of the manuscript.

**Competing interests:** The authors have declared that no competing interests exist.

movement, sensory, and speech impairments. 31 studies included prevalence of gross neurologic deficits at hospital discharge (cerebral malaria (CM), n = 26, broader forms of severe malaria, n = 5). Prevalence of deficits at hospital discharge in children with CM was 15.2% (95%CI: 11.5–18.8) ($I^2$ = 89.1%), compared to 2.4% (95%CI: 2.0–2.8) ($I^2$ = 36.6%) in children with broader forms of severe malaria. Prevalence of deficits in CM decreased from 15.2% to 5.7% (95%CI: 1.2–10.2) ($I^2$ = 79.2%) after 12 months follow-up. At regional level, Sub-Saharan Africa had a prevalence of 14.2% (95%CI: 10.5–17.9) ($I^2$ = 94.9%) compared to East Asia & Pacific at 2.3% (95%CI: 0.8–3.8) ($I^2$ = 0.0%) and South Asia at 6.1% (95%CI: 0.0–14.8) ($I^2$ = 0.0%). Risk factors for gross neurologic deficits included profound coma, coma lasting ≥48 hours, multiple convulsions, hypoglycaemia, and acute kidney injury.

## Conclusions

Gross neurologic deficits are more prevalent in children in Sub-Saharan Africa with CM compared to severe malaria in general, and a number of clinical factors in children with CM increase the risk. Interventions by clinicians should target children with CM at highest risk during admission.

## Introduction

Gross neurologic deficit (GND) refers to an abnormal neurologic function of a specific body part arising from injury to the brain, spinal cord, muscles or nerves that feed the affected area [1]. This may manifest in the form of a motor deficit (spasticity, cranial nerve palsies and central hypotonia), movement disorder (ataxia, tremors and dystonia), sensory deficit (visual or hearing) or speech and language impairments (aphasia) [2]. Severe malaria, most notably cerebral malaria (CM), has been implicated as a leading cause of brain injury [3] with about 11% of children surviving CM having GND at hospital discharge [2].

Some GND in children with severe malaria may be transient, while others may be long term evidenced by a number of studies that have reported marked declines in the prevalence of GND a few to many months following hospital discharge, and persistence in a subset of individuals [4–9]. Some deficits such as cortical blindness are generally transient while others such as plegia, paresis, extrapyramidal features, and epilepsies persist [1].

A number of factors have been reported to increase the risk for neurological sequelae following CM among children. These include deep and prolonged coma, multiple seizures, recurrent hypoglycaemia, acute kidney injury, anaemia, intracranial hypertension, age younger than 3years, male sex, and a biphasic clinical course marked by recovery of consciousness followed by recurrence of convulsions and coma [3,5,10–17]. However some recent studies have not confirmed the importance of previously associated factors such as anaemia, age, sex, multiplicity of convulsions, and persistence of GND after CM [1]. In addition, no systematic review has synthesized evidence on the prevalence or risk factors of GND among children after severe malaria [1].

Given that CM is a leading cause of brain injury, knowledge on important risk factors and prevalence of GND after SM may act as a guide to policy makers and clinicians in developing and targeting interventions to those most at risk, to prevent and control these deficits. We therefore performed a systematic review and meta-analysis to synthesize evidence on the prevalence and risk factors for GND after SM in childhood.

## Methods

The systematic review and meta-analysis was done following a protocol which was registered in the International Prospective Register of Systematic Reviews (CRD42022297109) and published [1] (S1 Protocol). We conducted the review in accordance with the Preferred Reporting Items for Systematic Reviews and Meta-Analyses (PRISMA) statement [18] (S1 Checklist).

### Search strategy

An experienced librarian and information specialist (AAK) searched MEDLINE, Embase, Web of Science, and Global Index Medicus (GIM) from the earliest date to 2nd February 2025 to identify studies that reported on GND after severe malaria in childhood. The search was performed using a combination of search terms, their synonyms and MeSH (Medical Subject Headings) whose details are provided in the published protocol [1] (S1 Protocol). The terms were combined using the Boolean operators 'AND' and 'OR'. We searched the bibliographies of all included studies for additional eligible studies. There were no language restrictions. Google Translate was used to translate eligible articles from French to English.

### Eligibility criteria

The review included all published studies from the earliest date that reported on prevalence and/or risk factors of GND after WHO-defined severe malaria in children (persons younger than 18 years). Study designs included: cross-sectional, case-control, cohort, randomized controlled trials, quasi-experimental study, or case series. The study setting included all malaria affected regions globally. Details of the eligibility criteria are in the published protocol [1] (S1 Protocol).

### Selection process and data management

EndNote software was utilized to initially organize references obtained from the search results. Duplicate entries were removed after importing the articles into EndNote v20. Following this, the studies were screened in duplicate using predefined eligibility criteria. The review team (AEO, SO, KOO, & CO) independently screened the articles in pairs using a pre-designed and pilot-tested screening tool within EPPI-Reviewer v6.16.0.0. Any conflicts during the screening process were resolved through consensus, with a third reviewer acting as a tiebreaker when needed.

### Data abstraction

A data abstraction tool was developed in EPPI-Reviewer which included key variables for abstraction from each article, namely: study characteristics (author, year of publication, year of data collection, title, citation, institution, country, language, source of funding), study design, population characteristics (sample size, age at diagnosis of severe malaria), type of gross neurologic deficit (including author specific definitions), duration of gross neurologic deficit, type of assessment for gross neurologic deficit, risk factors, measures of association (such as risk ratios, odds ratios), and prevalence of gross neurologic deficit [1] (S1 Protocol). The tool was initially tested on 5% of the eligible studies to confirm its effectiveness in capturing all necessary information before being uploaded to EPPI-Reviewer v6.16.0.0. The four reviewers (AEO, SO, KOO, & CO) independently abstracted the data, and any disagreements were resolved through discussion. Coding was independently performed by pairs of research team members and again, any discrepancies addressed through discussion and consensus. The finalized data was then reviewed by an independent senior reviewer to verify accuracy and ensure quality control and assurance.

 

## Risk of bias assessment

Risk of bias assessment was carried out using ROBINS-E ('Risk Of Bias In Non-randomized Studies- of Exposures') tool for non-randomized controlled trials (https://www.riskofbias.info/welcome/robins-e-tool) by the four reviewers (AEO, SO, KOO, & CO) in duplicate. This is a domain-based tool and risk of bias in included studies was assessed as a judgment (low risk of bias, some concerns, high risk of bias, very high risk of bias) using seven domains. These domains are risk of bias due to confounding; risk of bias arising from the measurement of the exposure; risk of bias in the selection of participants into the study (or into the analysis); risk of bias due to post-exposure interventions; risk of bias due to missing data; risk of bias arising from the measurement of the outcome; risk of bias in the selection of the reported result.

## Data analysis and synthesis

The pooled prevalence with 95% confidence of gross neurological deficits (GND) was analysed as the outcome in this study. Effect sizes were statistically pooled using Restricted Maximum Likelihood (REML) Random Effects Meta-analysis in R version 4.5.0 [19]. REML Random Effects Meta-Analysis uses restricted (or residual) maximum likelihood, which adjusts for the degrees of freedom lost by estimating fixed effects and provides a less biased and typically more accurate estimate of heterogeneity, this was applied because heterogeneity was moderate to high. The individual studies included in this review were drawn from different populations and contexts; hence, the random effects model was used to calculate the pooled mean effect size (proportion). These effects models are usually recommended when collecting data from a series of studies where the effect size is expected to vary from one to the next, and studies are unlikely to be functionally equivalent [20], as was the case for this review, which included studies from across the globe. Furthermore, random effects models enable statistical inferences to be made to a population of studies other than those included in the meta-analysis.

The pooled effect size was transformed using Freeman–Tukey double arcsine transformation to stabilize the variance between studies and was back transformed and expressed as a raw proportion for presentation. In studies where multiple effect sizes were reported from the same sample the mean of the combined effect sizes was calculated. In cases where studies used overlapping samples, an overall estimate was calculated and those that report effect sizes from independent subgroups, each subgroup was treated as a separate sample in the meta-analysis.

The synthesis is presented in form of a summary of findings tables, simple graphs, and forest plots, as appropriate. This followed the format of the Cochrane Consumer and Communication Review Group [21]. We described the included articles, organized and tabulated the results to identify patterns and converted the results into a common descriptive format. We undertook the form of outcome data tables, simple graphs, and forest plots, as appropriate. These was incorporated into the summary of findings tables, which informed the syntheses for dissemination hence we used both narrative and quantitative synthesis. Since data on each of the risk factors reported was not adequate to pool the effect sizes, we performed a narrative synthesis without meta-analysis (SWiM) [22] using the direction of effect approach.

## Assessment of heterogeneity

The level of heterogeneity in the articles was established using $I^2$ statistic. The $I^2$ statistic was displayed as the percentage (%) of heterogeneity due to between-study variation [23]. The value of $I^2$ statistic ≤ 25%, ≤ 50%, ≤ 75% ≥ 75% indicated low, medium, substantial and high heterogeneity among studies, respectively. Increase in heterogeneity implies underlying contextual differences between studies and limits generalizability. Subgroup analysis was performed on articles with substantial to high heterogeneity [24].

## Sub-group & sensitivity analysis

The sensitivity analysis was done by removing studies from the meta- analysis one-by-one to see if the results of the meta-analysis are sensitive to any single study. We also examined sensitivity of findings to risk of bias status (good, fair

 

and poor quality). Additional sub-group analyses were performed to compare prevalence of GND by: duration of follow-up, that is, at discharge, three, six, 12 and over 12-months follow-up; by region, that is, Sub-Saharan Africa, East Asia & Pacific, and South Asia region, since studies were found to report on GNDs at different follow-up time-points and in different regions; by study design; and by type of treatment.

## Publication bias

We assessed the risk of publication bias in the included articles by using the asymmetry of funnel plots [25]. This technique uses ranking methods for data augmentation and has demonstrated reliability in evaluating publication bias caused by missing studies or data. A funnel plot was employed to visually examine the symmetry of effect sizes, aiding in the detection of potential bias, particularly when smaller studies tend to report more pronounced effects. In the absence of missing studies or small study effects, the scatter plot looks like a symmetrical inverted funnel with a wide base and a narrow top [25]. The presence of large "holes" or asymmetry in the plot suggests publication bias, however this might also be explained by other factors such as study heterogeneity.

## Dealing with missing data

In case of missing data from the published articles, study authors were contacted. When the author could not be accessed or in case of no response from authors, we reported the characteristics of the study but did not include such a study in the meta-analysis.

## Deviations from the protocol

The effect size used was a raw proportion of children who developed GNDs as opposed to the Freeman-Tukey double arcsine transformation approach as stated in the protocol. We used the random effects model because this method accounts for heterogeneity between studies, which is expected in this study due to variations in demographics and study design, and it assumes that the true effect could vary from study to study [26]. While the fixed effects model assumes that one true effect size underlies all studies in a meta-analysis and that any differences are due to sampling error [26].

We compared the prevalence of GND among children in studies specifically enrolling children with CM compared to studies with broader forms of SM (including CM), which is a deviation from the protocol that indicated that we would compare the prevalence of GND in studies with CM compared to studies with non-CM. This was because studies that reported GND in different forms of SM did not disaggregate them by SM criteria, therefore we could not determine the prevalence of GND in the non-CM group.

We did not perform a meta-analysis for different age-groups as stated in the protocol because most of the studies did not disaggregate prevalence of GND by age-group and there were variations in the age groups reported.

## Results

### Study selection

An initial search of databases namely, MEDLINE, Embase, Web of Science, and Global Index Medicus from 1946 up to 11 June 2023, yielded 2856 potentially relevant publications (MEDLINE = 750, EMBASE = 1249, Web of Science = 847, GIM = 10). An updated search was conducted up to 2nd February, 2025, which yielded a total of 264 potentially relevant publications (MEDLINE = 77, EMBASE = 155, Web of Science = 22, GIM = 10). Of the total records identified from both searches (3120), 889 were excluded after duplicate removal, resulting in 2231 records for title and abstract screening (Fig 1). A total of 2000 records were excluded after title and abstract screening, resulting in 231 articles. Full-texts of twenty-six articles were not retrieved, resulting in 205 articles for full-text screening. After full-text screening, 174 did not meet the eligibility criteria and were excluded. An additional 10 studies that met the eligibility criteria were identified

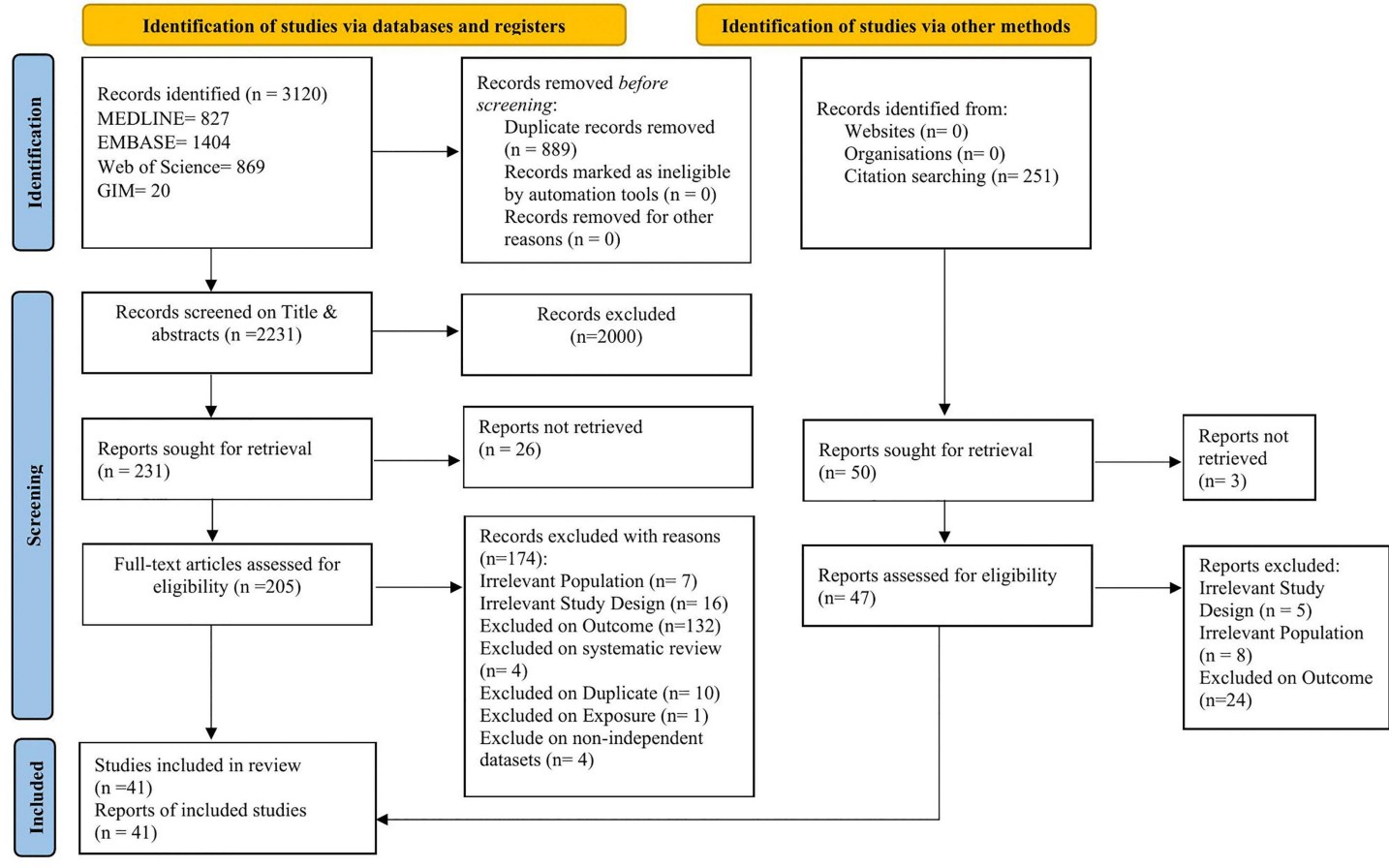

**Fig 1. PRISMA flow chart showing the study selection process.**

through searching the bibliography of the included studies. A total of forty-one studies met the eligibility criteria and were included in the systematic review (Fig 1).

## Characteristics of included studies

Among the forty-one studies, thirty-four reported on GND after cerebral malaria (CM), while seven reported on GND after different forms of severe malaria. These studies were from 16 countries in three regions, namely: i) Sub-Saharan Africa (Uganda, The Gambia, Kenya, Nigeria, Malawi, Togo, Mozambique, Ghana, Tanzania, Rwanda, Democratic Republic of Congo, South Sudan, Mali); ii) East Asia & Pacific (Papua New Guinea); and iii) South Asia (India, Pakistan) (Table 1). The types of GND included: motor deficits such as paresis, plegia, hypertonia, hypotonia, dystonia, divergent squint, oral dyspraxia; movement deficits such as ataxia; sensory deficits described as deafness, impaired hearing, cortical blindness, visual impairment; speech impairments described as delayed speech, and dysphasia; and a vegetative state. Thirty-one studies reported GND at discharge, two studies reported GND at three months, seven studies reported GND at six months, three studies reported GND at 12-months, and five studies reported GND after 12 months. Nine studies reported adjusted odds ratios (AOR) for the association between risk factors and GND (Table 2). Study designs included, prospective cohort (n = 32), retrospective cohort (n = 3), Randomized-Controlled Trials (n = 4), and cross-sectional (n = 2). Among studies that reported on treatment, types of treatment included: chloroquine (IV or subcutaneous) (n = 2), chloroquine or IV

**Table 1. Summary of included studies on prevalence of gross neurologic deficits after childhood severe malaria.**

| Study (Year) | Country | Type of SM | Study Design | Type of Gross Neurologic Deficit | Prevalence at Discharge n/N (%) | Prevalence after discharge n/N (%) |
|---|---|---|---|---|---|---|
| Schmutzhard et al (1984) [27] | Tanzania | CM | Prospective cohort | Hemiparesis | NR | 4/49 (8.2%) at 6 months |
| Ahmad et al (1986) [28] | India | CM | Prospective cohort | Hemiparesis, monoparesis, isolated cranial nerve palsy (ptosis and facial) | 1/20 (5%) | |
| Molyneux et al (1989) [4] | Malawi | CM | Prospective cohort | Hemiparesis, generalized hypotonia, generalized spasticity of limbs, cerebellar ataxia, bilateral 3–4/s extrapyramidal tremor | 12/111 (10.8%) | 7/111 (6.3%) at 1 month |
| Brewster et al (1990) [5] | The Gambia | CM | Prospective cohort | Hemiplegia, cortical blindness, aphasia, ataxia, generalised spasticity, tremors | 32/265 (12.1%) | 6/265 (2.3%) at 6 months |
| Bondi et al (1992) [10] | Nigeria | CM | Prospective cohort | Monoparesis, cortical blindness, hearing loss, dysphasia, dysarthria, spastic quadriplegia, hemiparesis, hemiplegia | 7/62 (11.3%) | |
| Schapira et al (1993) [29] | Mozambique | CM | Randomized Controlled Trial | Cortical blindness, inferior paraparesis | 4/54 (4.4%) | |
| Walker et al (1993) [30] | Nigeria | CM | Randomized Controlled Trial | Motor weakness, cortical blindness, loss of neurological milestones | 5/45 (11.1%) | |
| Steele et al (1995) [31] | Ghana | CM | Retrospective cohort | Cortical blindness, decreased vision, hypotonia, spastic hemiplegia, deafness, aphasia, gait disturbance, hemiplegia, facial palsy | 42/187 (22%) | |
| Newton et al (1996) [32] | Kenya | CM | Prospective cohort | Hemiparesis, blindness, hand dystonia, divergent squint, mild diplegia, vegetative state | 11/44 (25%) | |
| Crawley et al (1996) [33] | Kenya | CM | Prospective cohort | Hemiplegia, spastic quadriplegia | NR | 7/58 (12%) at 3 months |
| Allen et al (1996) [34] | Papua New Guinea | CM | Prospective cohort | Severe spastic quadriparesis, severe spastic quadriplegia, poor co-ordination of the left arm and hand, mild spasticity in both legs | NR | 4/66 (6%) at 6 months |
| Genton et al (1997) [35] | Papua New Guinea | CM | Prospective cohort | Right hemiplegia, involuntary movements | 2/118 (1.7%) | 2/50 (4%) at 1 month |
| Van Hensbroek et al (1997) [11] | The Gambia | CM | Prospective cohort | Paresis, ataxia, hearing defects, visual field defects, aphasia, developmental regression | NR | 29/466 (6.2%) at 1 month, 18/452 (4.0%) at 6 months |
| Assimadi et al (1998) [36] | Togo | SM | Prospective cohort | Aphasia, hemiplegia, oculomotor paralysis, cerebellar ataxia | 9/445 (2%) | |
| Olumese et al (1999) [6] | Nigeria | CM | Prospective cohort | Cortical deafness, abnormal gait, aphasia, right hemiparesis, severe hypotonia and other motor deficits | 12/103 (11.7%) | 1/103 (1%) at 1 month |
| Crawley et al (2001) [37] | Kenya | CM | Prospective cohort | Hemiplegia, spastic quadriplegia | 6/58 (10.3%) | |
| Oguche et al (2002) [7] | Nigeria | CM | Prospective cohort | Spastic quadriparesis/aphasia, motor deficits/regression of motor milestone, hearing impairment/abnormal behavior | 5/39 (12.8%) | 1/39 (2.6%) at 1 month |
| Idro et al (2004) [12] | Uganda | CM | Prospective cohort | Cortical blindness, impaired hearing, spastic quadriparesis, aphasia, generalised hypotonia | 3/93 (3.2%) | |

*(Continued)*

**Table 1.** (Continued)

| Study (Year) | Country | Type of SM | Study Design | Type of Gross Neurologic Deficit | Prevalence at Discharge n/N (%) | Prevalence after discharge n/N (%) |
|---|---|---|---|---|---|---|
| Idro et al (2005) [38] | Kenya | CM | Cross-sectional | Motor impairments (central hypotonia, paresis, ataxia), aphasia, blindness, deafness | 39/354 (11%) | |
| Idro et al (2006) [3] | Kenya | CM | Prospective cohort | Motor impairments, speech & language impairments, hearing & visual impairment | NR | 22/143 (15.4%) median 64 months (IQR 40–78) |
| Ngoungou et al (2006) [39] | Mali | CM | Prospective cohort | Speech delay, oral dyspraxia, diplegia, dystonia | NR | 5/101 (5%) at about 36 months |
| Medana et al (2007) [40] | Kenya | CM, PR, MS | Prospective cohort | Quadriparesis, generalized hypotonia | 3/117 (2.6%) | |
| Ahmed et al (2008) [41] | Pakistan | CM | Retrospective cohort | Motor deficit, cranial nerve palsy | 1/9 (11.0%) | |
| Casals-Pascual et al (2008) [42] | Kenya | CM | Retrospective cohort | Visual impairment, impairment of speech, motor impairment (hemiparesis, quadriparesis, monoparesis) | 32/108 (30%) | |
| Opoka et al (2009) [8] | Uganda | CM | Prospective cohort | Hyperreflexia and hypertonia, spastic quadriplegia, vision and hearing impairments, ataxia, lack of coordination and attention deficit | 19/76 (25.0%) | 7/76 (9.2%) at 3 months, 2/74 (2.7%) at 6 months, 1/68 (1.5%) at 24 months |
| Birbeck et al (2010) [13] | Malawi | CM | Prospective cohort | Hemiparesis, monoparesis, hypotonia, gait ataxia, speech regression, choreiform movements, spastic quadriparesis, language regression, cortical blindness | 12/132 (13.6%) | |
| Dondorp et al (2010) [43] | Mozambique, The Gambia, Ghana, Kenya, Tanzania, Nigeria, Uganda, Rwanda, D.R.C | SM | Randomized Controlled Trial | Severe motor impairment, cortical blindness, severe speech and hearing impairment | 115/4898 (2.3%) | 191/507 (37.7%) at 28 days~1 month |
| Bhanushali et al (2011) [44] | Malawi | CM | Prospective cohort | Motor deficits, sensory deficits, ataxia | NR | 4/25 (16%) at 12 months |
| Manning et al (2011) [45] | Papua New Guinea | CM, SMA, RD, MS, PR | Prospective cohort | Cortical blindness, motor deficits (ranging from mild ataxia to spastic quadriparesis) | 7/261 (2.7%) | |
| Oluwayemi et al (2013a) [46] | Nigeria | CM | Prospective cohort | Ataxia, speech impairment, cortical blindness, hearing impairment, hemiparesis, spastic quadriplegia, spastic triplegia, quadriparesis, left-sided ptosis | 22/57 (38.6%) | |
| Oluwayemi et al (2013b) [47] | Nigeria | CM | Prospective cohort | Visual impairment, speech impairment, monoparesis, quadriparesis, hearing impairment, movement disorder | 16/131 (12%) | |
| Mergani et al (2015) [48] | South Sudan | CM | Cross-sectional | Hemiparesis, blindness, hemiplegia, aphasia, quadriplegia, quadriparesis | 64/351 (18%) | |
| Hawkes et al (2015) [49] | Uganda | All SM | Randomized controlled trial | Inability to sit, spastic or flaccid paresis of one or more limbs, seizures, unilateral weakness, vision loss, gaze palsy, and poor head control | 12/164 (7.3%) | |
| Oninia et al (2015) [50] | Nigeria | CM | Retrospective cohort | Motor deficit, cortical blindness, aphasia, hearing impairment | 6/17 (35.3%) | |
| O'Brien et al (2018) [51] | Democratic Republic of Congo | CM | Prospective cohort | Aphasia, ataxia, central hypotonia, choreiform movements, hemiparesis, blindness | 14/121 (11.6%) | 14/121 (11.6%) at 1 month |

*(Continued)*

**Table 1.** (Continued)

| Study (Year) | Country | Type of SM | Study Design | Type of Gross Neurologic Deficit | Prevalence at Discharge n/N (%) | Prevalence after discharge n/N (%) |
|---|---|---|---|---|---|---|
| Conroy et al (2021) [52] | Uganda | CM | Prospective cohort | Motor deficits, ataxia, movement disorders, speech disorders, visual disorders | 83/232 (35.8%) | 11/223 (4.9%) at 6 months, 7/223 (3.1%) at 12 months, 6/220 (2.7%) at 24 months |
| Alabi et al (2023) [14] | Nigeria | CM | Prospective cohort | Abnormal posturing, absent cornea reflexes, abnormal pupillary light reflexes, hypotonia, hypertonia, depressed tendon reflexes, hyper-reflexia | 11/41 (26.8%) | |
| Chastang et al (2023) [15] | Malawi | CM | Prospective cohort | Abnormal hearing, movement, vision, tone, development | 152/1395 (10.9%) | |
| Prasad (2023) et al [53] | India | CM | Prospective cohort | Motor deficit, visual impairment | | 7/35 (20%) at 6 months |

quinine (n = 2), IV quinine (n = 6), ACTs (IV artesunate or IM Artemether) or IV quinine (n = 8) and ACTs (n = 4). The review had a total of 11,635 children with ages ranging from one month to 15yrs. A list of the excluded studies at full text along with the reasons for exclusion has been provided as an additional file, S1 Table.

### Risk of bias in the included studies

Eighteen of the included studies (18/41: 43.9%) had low risk of bias, seventeen (17/41: 41.5%) had a moderate risk of bias while six (6/41: 14.6%) had high risk of bias (S1 Fig). Following the risk of bias assessment criteria used, included studies had potential risk of: bias due to confounding (12/41: 29.3%), bias due to post-exposure interventions (9/41: 22%), bias due to missing data (3/41: 7.3%), bias arising from measurement of the outcome (1/41: 2.4%), and bias in selection of the reported result (12/41: 29.3%) (S1 and S2 Figs).

### Publication bias

We used the funnel plot and Egger's test to assess for publication bias. The funnel plot was asymmetrical which indicated presence of publication bias (S3 Fig). The Egger's test was statistically significant (p < 0.0001) which strongly suggested evidence of small-study effects an indication of publication bias as shown in the funnel plot. The bias estimate 4.05 (Standard Error = 0.64) represented the intercept of the regression line used in Egger's test. A value far from zero (and statistically significant) which implied that the funnel plot is asymmetric. This positive value of the bias estimate suggested that smaller studies are more likely to report higher proportions of gross neurological deficits, which may indicate that studies with low prevalence rates may be underreported or unpublished.

### Prevalence of gross neurologic deficits in children with severe malaria at hospital discharge globally

A total of 10,108 participants were included from 31 different studies to determine the prevalence of GND at hospital discharge (Fig 2). Of these, 26 studies reported GND after CM while five reported GND after different forms of SM (including CM). Among participants who only had CM, the prevalence of GND was 15.2% (95%CI: 11.5–18.8) ($I^2$ = 89.1%) while among those who had broader forms of SM (including CM), the prevalence of GND was 2.4% (95%CI: 2.0–2.8) ($I^2$ = 36.6%) (Fig 2). Overall, the prevalence of participants with GND after severe malaria at hospital discharge was 13.0% (95%CI: 9.5–16.4), however there was a high heterogeneity in the included studies ($I^2$ = 94.2%) between the two subgroups (Fig 2).

**Table 2. Summary of included studies on risk factors of gross neurologic deficits after childhood severe malaria.**

| Study (Year) | Country | Type of SM | Duration of follow-up | Risk Factors for Gross Neurologic Deficits | OR (95% CI) | AOR (95% CI) |
|---|---|---|---|---|---|---|
| Van Hensbroek et al (1997) [11] | The Gambia | CM | 6 months | Coma score 0 or 1<br>Duration of coma>=48hrs<br>Multiple convulsions | 6.9 (1.9-25.3)<br>11.2 (3.0-42.4)<br>6.0 (2.0-18.3) | 7.4 (1.8-29.7)<br>13.4 (3.4-52.4)<br>7.1 (2.2-22.7) |
| Oguche et al (2002) [7] | Nigeria | CM | 1 month | Metabolic acidosis<br>Elevated plasma creatinine | NR<br>NR | 4.5 CI NR<br>10.5 CI NR |
| Idro et al (2004) [12] | Uganda | CM | Hospital admission | Multiple convulsions | NR | 12.8 (3.0–211) |
| Idro et al (2006) [3] | Kenya | CM | 64 months | Previous admissions<br>Previous history of seizures<br>Age<3 years<br>Hypoglycemia at or during admission<br>Features of raised intracranial pressure<br>Multiple (3 or more seizures)<br>Focal neurologic signs<br>Deep coma<br>Severe malnutrition | NR<br>NR<br>NR<br>NR<br>NR<br>NR<br>NR<br>NR<br>NR | 4.8 (1.3-17.6)<br>7.6 (1.7-33.6)<br>6.4 (1.2-36)<br>6.1 (1.4-25.7)<br>6.0 (1.4-30.6)<br>8.3 (2.3-29.5)<br>12.3 (1.4-110)<br>28.8 (3.0-280)<br>6.6 (1.4-30.6) |
| Casals-Pascual et al (2008) [42] | Kenya | CM | Hospital admission | Admission with profound coma<br>Convulsions | 7.7 (2.5-24.0)<br>3.4 (1.1-10.5) | 5.5 (1.5-20.7)<br>16.4 (2.9-90.8) |
| Mergani et al (2015) [48] | South Sudan | CM | Hospital admission | Abnormal posture either decerebration or decortication<br>Focal convulsions<br>Coma >48 hours | 6.1 (3.3-11.0)<br>0.6 (3.6-30.7)<br>4.1 (2.1-8.0) | 5.3 (2.7-10.6)<br>11.3 (5.5-23.2)<br>3.8 (1.9-7.9) |
| Conroy et al (2019) [17] | Uganda | CM & SMA | 24 months | Acute kidney injury | NR | 3.0 (1.2-7.6) |
| Namazzi et al (2022) [16] | Uganda | CM, SMA, MS, PR, RDS | Hospital admission | Acute kidney injury<br>Unresolved acute kidney injury<br>Elevated BUN (>20mg/dL) | 2.4 (1.5-4.0)<br>4.6 (2.4-8.7)<br>2.8 (1.7-4.8) | 2.0 (1.2-3.4)<br>3.7 (1.9-7.2)<br>2.8 (1.7-4.9) |
| Chastang et al (2023) [15] | Malawi | CM | Hospital admission | Hypoglycemia at admission | 2.7 | 3.2 (1.5-6.9) |

## Sensitivity analysis

By removing studies one by one and restricting the analysis among studies with low and moderate risk of bias (RoB), the pooled prevalence of GND among children with CM was 12.9% (95%CI:9.0–16.8) ($I^2$=94.7%) (S4 Fig) which was not statistically different from the overall pooled prevalence of GND among children with CM at 13% (95%CI: 9.5–16.4) ($I^2$=94.2%) (S5 Fig).

## Prevalence of gross neurologic deficits among children with cerebral malaria at 3, 6, 12 and >12 months follow-up globally

A total of 12 studies provided data for prevalence of GND after hospital discharge (Figs 3–6).

At three months follow-up, the prevalence of GND among 134 children with CM was 10.3% (95%CI: 5.2–15.4), ($I^2$=0.0%) (Fig 3).

Prevalence of post-discharge GND among children with CM was 3.9% (95%CI: 2.5–5.2) among studies that reported GND at six months follow-up ($I^2$=47.0%) (Fig 4).

At 12 and over 12 months follow-up, prevalence of GND among children with CM was not statistically different at 5.1% (95%CI: 0.9–9.4) ($I^2$=47.5%) (Fig 5), and 5.7% (95%CI: 1.2–10.2) ($I^2$=79.2%) (Fig 6) respectively among studies that reported GND in these periods.

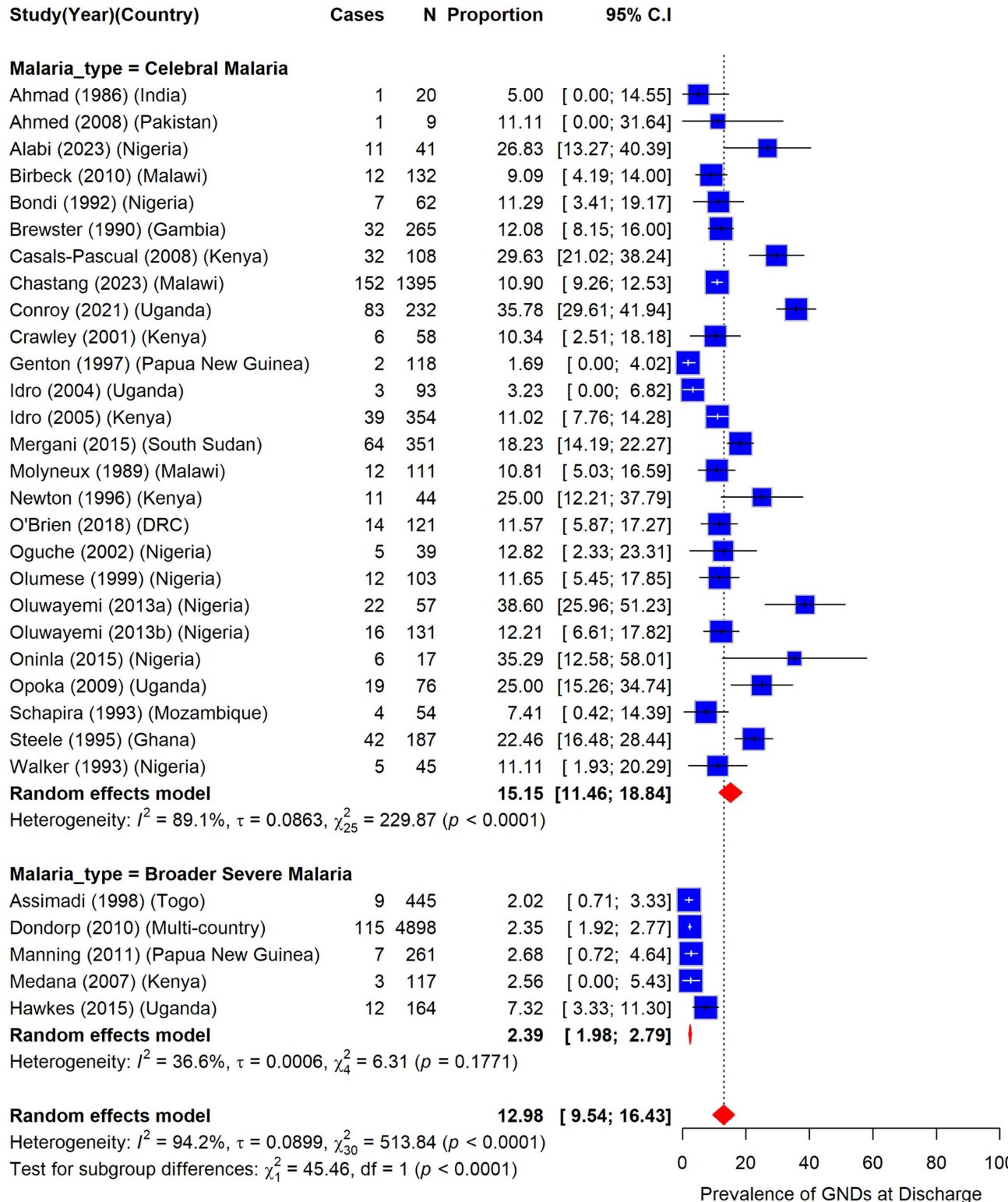

**Fig 2. Prevalence of gross neurologic deficits among children with severe malaria at hospital discharge.**

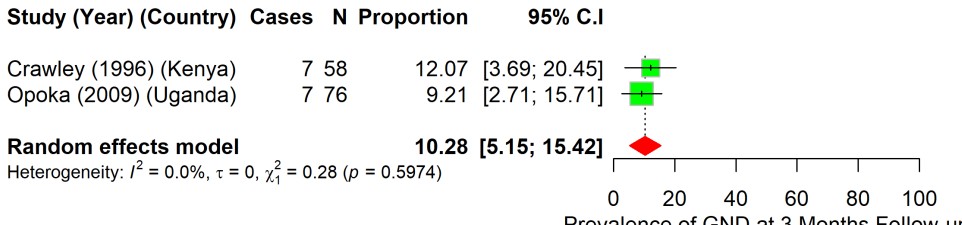

**Fig 3. Prevalence of gross neurologic deficits among children who had CM at 3-month follow-up.**

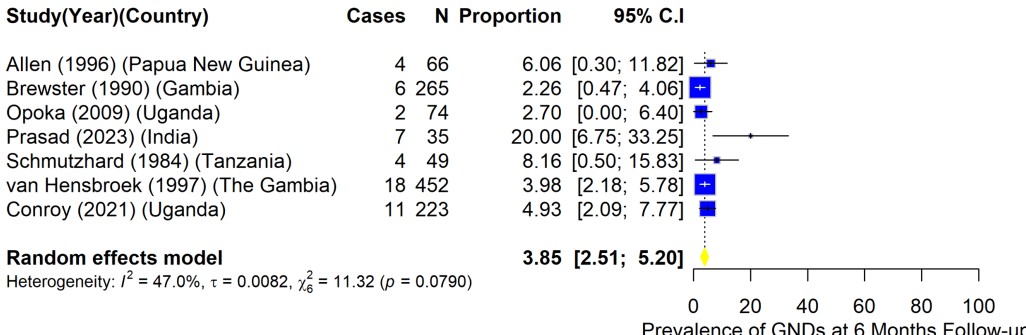

**Fig 4. Prevalence of gross neurologic deficits among children who had CM at 6-month follow-up.**

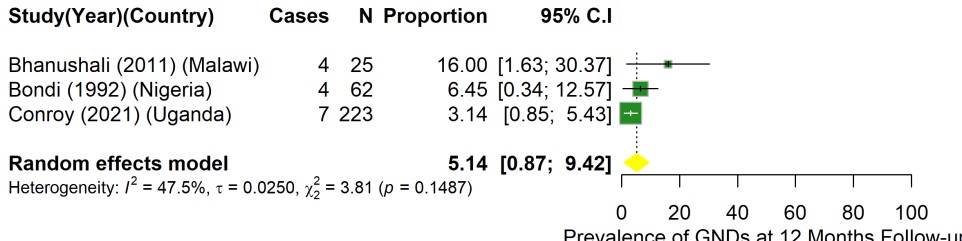

**Fig 5. Prevalence of gross neurologic deficits among children who had CM at 12-month follow-up.**

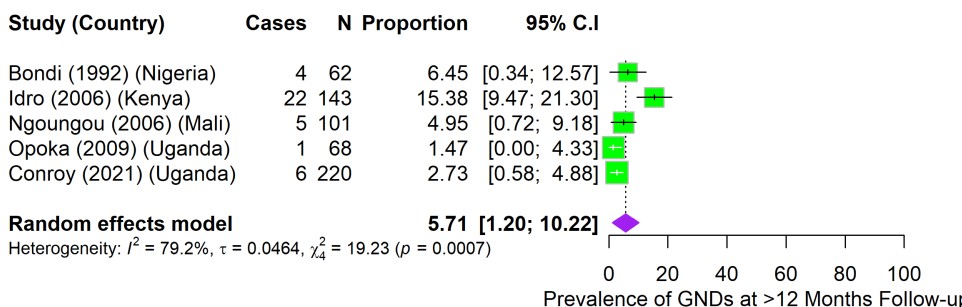

**Fig 6. Prevalence of gross neurologic deficits among children who had CM at over 12-month follow-up.**

### Prevalence of gross neurologic deficits among children with severe malaria by region

Sub-Saharan Africa accounted for a significantly higher prevalence of GND among participants with severe malaria at hospital discharge at 14.2% (95%CI: 10.5–17.9) ($I^2$ = 94.9%) compared to South Asia 6.1% (95%CI: 0.0–14.8) ($I^2$ = 0.0%) and East Asia & Pacific, 2.3% (95%CI: 0.8–3.8) ($I^2$ = 0.0%) (Fig 7).

### Prevalence of gross neurologic deficits among children with severe malaria by study design

Retrospective cohort study designs reported the highest prevalence of GND among study designs at hospital discharge at 24.8% (95%CI: 18.8–30.9) ($I^2$ = 30.4%) while randomized controlled trials reported the lowest prevalence at 5.5% (95%CI: 0.3–10.7) ($I^2$ = 63.5%). The remaining study designs, that is, the prospective cohort and cross-sectional studies had similar prevalences at 12.5% (95%CI: 8.2–16.8) ($I^2$ = 92.8%) and 12.2% (95%CI: 6.0–18.4) ($I^2$ = 86.5%) (Fig 8).

### Prevalence of gross neurologic deficits among children with severe malaria by type of treatment

The prevalence of GNDs among children in different treatment groups were as follows: chloroquine, 9.9% (95%CI: 3.6–16.3) ($I^2$ = 44.6%); quinine, 16.0% (95%CI: 5.8–26.1) ($I^2$ = 94.0%); chloroquine/quinine, 17.2% (95%CI: 6.3–28.1) ($I^2$ = 79.6%); ACTs/quinine, 14.9% (95%CI: 7.0–22.9) ($I^2$ = 97%); and ACTs, 10.3% (95%CI: 2.0–18.7) ($I^2$ = 86.1%). However, the difference in prevalence of GNDs by treatment type was not statistically significant (P = 0.67) ($I^2$ = 95%) (Fig 9).

### Risk factors for gross neurologic deficits in children with cerebral malaria globally

Nine studies reported adjusted odds ratios (AORs) for risk factors of GND following childhood cerebral malaria [3,7,11,12,15–17,42,48] (Table 2). Risk factors associated with GND at hospital discharge included: focal convulsion, AOR = 11.3, 95%CI: 5.5–23.2 [48]; unresolved acute kidney injury (AOR = 3.7, 95%CI: 1.9–7.2) along with elevated blood urea nitrogen (BUN) (>20mg/dL) (AOR = 2.8, 95%CI: 1.7–4.9) [16]; metabolic acidosis, AOR = 4.5, and elevated plasma creatinine, AOR = 10.5 [7], suggestive of acute kidney injury, which were associated with short-term deficits at 1 month. Some risk factors were associated with GND both at hospital discharge, and long-term (GND persisting >3months duration of follow-up). These included: admission with profound/deep coma, AOR = 5.5, 95%CI: 1.5–20.7 [42], AOR = 7.4, 95%CI: 1.8–29.7 [11] and AOR = 28.8, 95%CI: 3.0–280 [3]; coma lasting for ≥48hours, AOR = 13.4, 95%CI: 3.4–52.4 [11] and AOR = 3.8, 95%CI: 1.9–7.9 [48]; multiple convulsions, AOR = 7.1, 95%CI: 2.2–22.7 [11], AOR = 12.8, 95%CI: 3.0–211 [12], and AOR = 16.4, 95%CI: 2.9–90.8 [42]; hypoglycemia, AOR = 6.1, 95%CI: 1.4–25.7 [3] and AOR = 3.2, 95%CI: 1.5–6.9 [15] and acute kidney injury, AOR = 3.0, 95%CI: 1.2–7.6 [17] and AOR = 2.0, 95%CI:1.2–3.4 [16] as shown in Table 2. Risk factors associated with long-term GND included: previous admissions (AOR = 4.8, 95%CI: 1.3–17.6), focal neurological signs observed during admission (AOR = 12.3, 95%CI: 1.4–110), multiple seizures (AOR = 8.3, 95%CI: 2.3–29.5), previous history of seizures (AOR = 7.6, 95%CI: 1.7–33.6), age below 3years (AOR = 6.4, 95%CI: 1.2–36), severe malnutrition (AOR = 6.6, 95%CI: 1.4–30.6), and features of raised intracranial pressure (AOR = 6.0, 95%CI: 1.4–30.6)) [3].

## Discussion

The review shows that prevalence of GND at hospital discharge was higher in studies where children had CM compared to those where children had different forms of SM, including but not limited to CM. The prevalence of GND among children with CM decreased significantly overtime following hospital discharge. Sub-Saharan Africa had a significantly higher prevalence of GND compared to the two Asia regions and a number of clinical factors that increase the risk of GND among children with CM were reported across several studies.

Prevalence of GND at hospital-discharge in studies reporting children with CM was seven times higher than that in studies with different forms of SM. All studies reviewed reported marked declines in the prevalence of GND among children with CM within a few months post-hospital discharge [4–9,54], demonstrating that some deficits are transient and may be a marker of severe illness that resolves overtime. The prevalence was highest at hospital discharge and reduced over follow-up. At

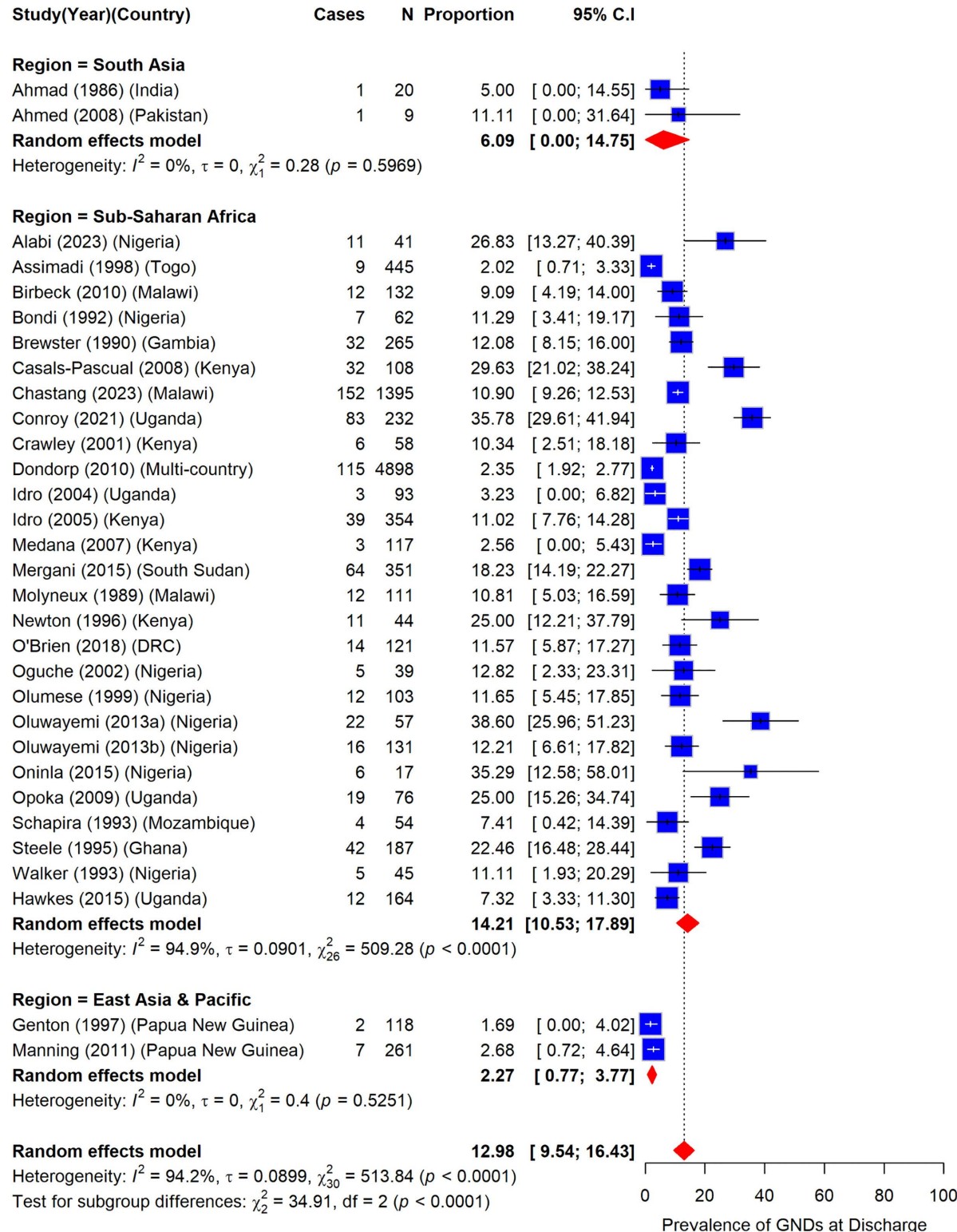

**Fig 7. Prevalence of gross neurologic deficits among children with severe malaria by region.**

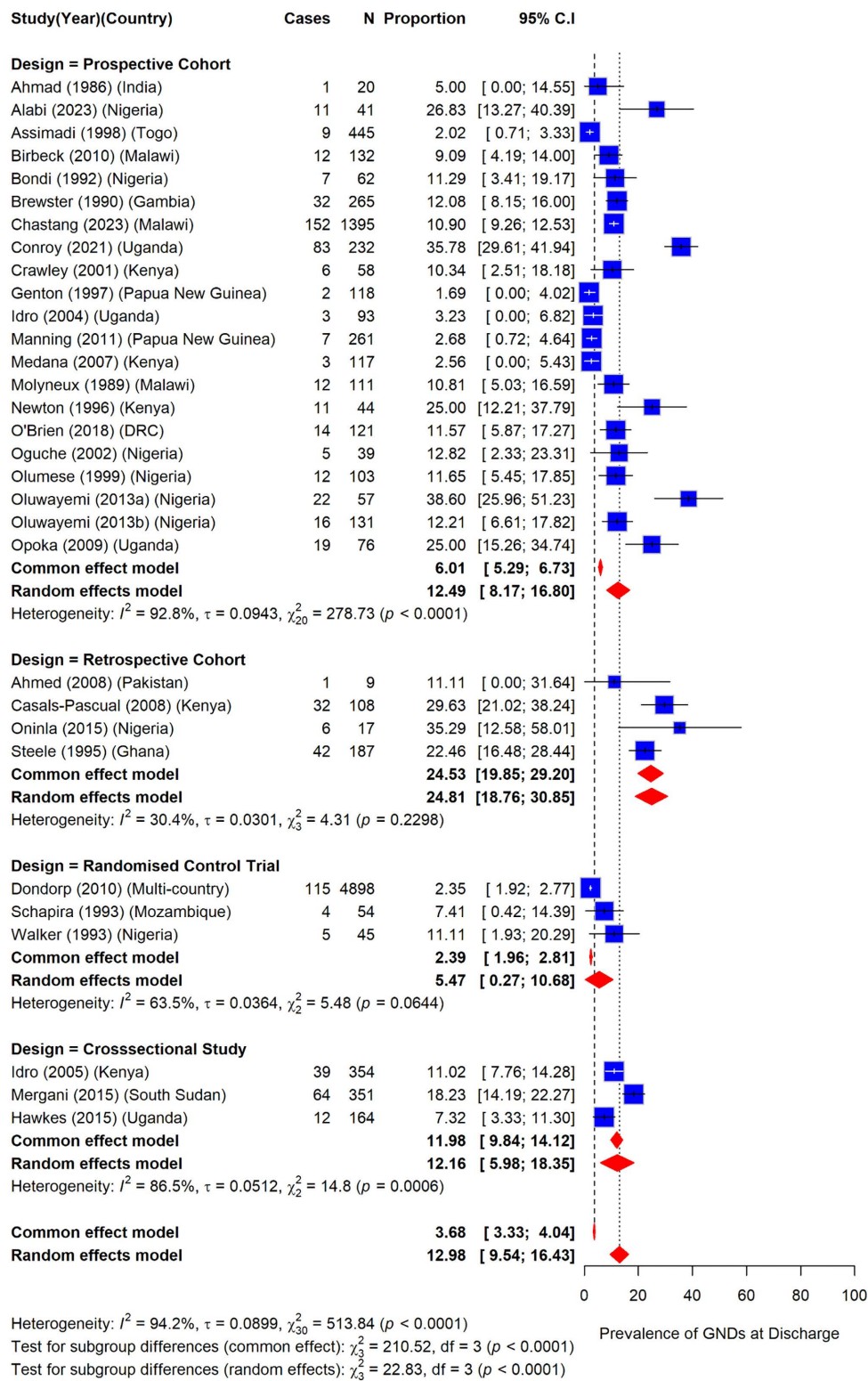

**Fig 8. Prevalence of gross neurologic deficits among children with severe malaria by study design.**

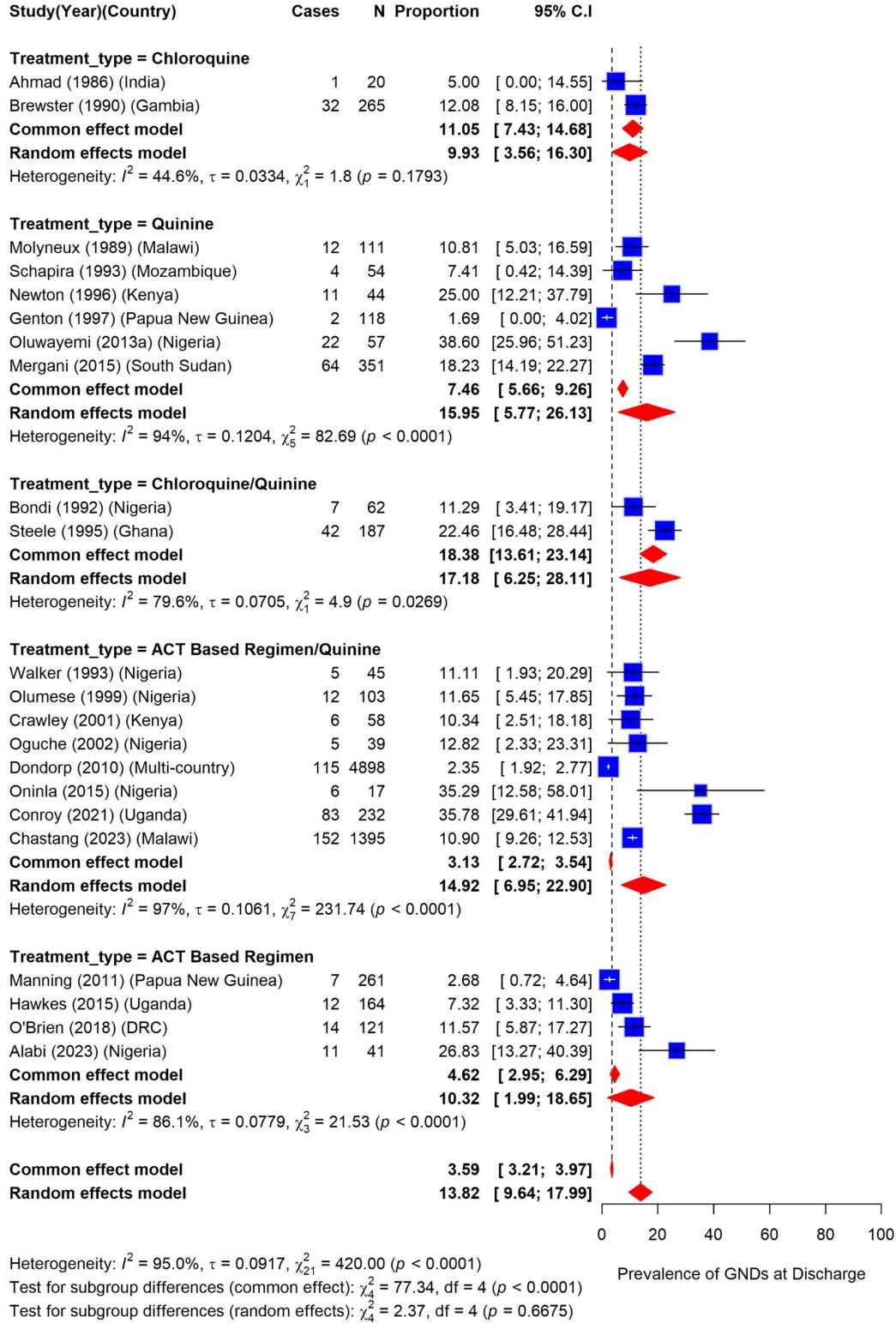

**Fig 9. Prevalence of gross neurologic deficits at hospital discharge by treatment type.**

three months post discharge, the pooled prevalence of GND reduced by over 30%. There was a further decline to six months after which the prevalence remained more-or-less stable beyond 12 months follow-up. This indicates that most recovery from GND after CM occurs within the first six months after acute illness [54] after which it is less likely to resolve.

Cerebral malaria is the most severe neurological complication of severe malaria [2], which has been associated with both mortality and neurological sequelae among children [2,14,54]. While malaria with multiple seizures has been associated with GND, the burden is much lower compared to children with CM [55]. Studies which had different forms of SM were predominantly populated by other forms of SM (such as SMA and malaria with multiple seizures) which could explain the significantly lower pooled prevalence of GND among children in this group compared to those who only had CM.

Factors such as admission with profound coma, coma lasting ≥48hours, hypoglycemia, features of raised intracranial pressure, severe malnutrition, metabolic acidosis, elevated plasma creatinine, acute kidney injury, unresolved acute kidney injury along with elevated blood urea nitrogen, convulsions, multiple seizures, previous history of seizures, focal neurologic signs observed during admission, previous admissions, and age below 3years were reported to increase the risk of GND after CM. The pathophysiology of CM involves the sequestration of red blood cells with parasitaemia in cerebral blood vessels which results in microvascular congestion, breakdown of the blood-brain barrier, and elevated intracranial pressure [56] which reduces cerebral perfusion pressure, nutrient and oxygen delivery [2] leading to herniation and obstruction of the blood vessels [56]. These processes may result in metabolic acidosis and elevated plasma creatinine as a result of acute kidney injury [17], and hypoglycaemia which may result in coma all of which have been implicated in neurologic sequelae [2]. It is likely that the more intense these processes leading to the coma, the deeper and more prolonged the coma, resulting into greater neurologic damage. Convulsions and seizures which may be attributed to excitation by quinolinic acid, buildup of parasitaemia as is common in childhood cerebral malaria and the resulting neuronal injury [2]. It is postulated that brain injury may be caused by the seizure initiating excitotoxic and other factors that may lead to a vicious cycle of neural injury and more seizures [2]. The reported age below 3 years as a risk factor for GND is consistent with a study in which the highest risk for motor, language and social development problems was among children under 5years [57] which suggests that GND and cognitive and social outcomes may both be worse in children with CM under 5years of age compared to older children [57].

The prevalence of GND in children with CM in Sub-Saharan Africa at hospital discharge was much higher than that in East Asia & Pacific, and South Asia regions. In a study conducted in Papua New Guinea in which the prevalence of GND among children with CM was particularly very low (1.7%), the proportion of children with hypoglycemia and deep coma, which is a marker of greater neurologic damage and a risk factor for GND [35], was notably much lower than that found in Sub-Saharan African countries [35]. A number of studies in Sub-Saharan Africa that have reported higher prevalences in GND among children with CM compared to that in reported in the East Asia & Pacific region, have also reported higher proportions of children exposed to these risk factors [5,10,11,42]. Thus, implying a correlation between the prevalence of GND and the proportion exposed to risk factors for GND, a likely reason for the comparatively higher prevalence of GND among children with CM in Sub-Saharan Africa compared to that in the East Asia & Pacific, and South Asia regions.

This review had some limitations. A number of studies did not indicate the type of examination that was used to assess GND, we therefore made an assumption that the standard neurologic examination was used which could have led to some reporting bias and might have had an effect on the outcome in the primary studies. Most of the studies did not report whether the examiners for neurologic sequelae were blinded to the exposure status of the study participants, we therefore cannot assess whether there could have been reporting bias which could have affected the outcomes reported in the primary studies. We limited the impact of this bias by conducting a risk of bias assessment and carrying out a meta-analysis on the prevalence of gross neurologic deficits among children with CM at discharge in studies with low and moderate risk of bias. The strength of this review is that it has addressed all the set objectives. We synthesized evidence on the prevalence of GND in children with severe malaria globally (in Sub-Saharan Africa, East Asia & Pacific, and South Asia), and over different periods of follow-up. We also synthesized evidence on the risk factors of GND after SM in childhood.

## Conclusions

One in every six children with CM has a GND at hospital discharge compared to one in every 50 children with different forms of SM. The burden of GND is also significantly higher in Sub-Saharan Africa compared to other regions, namely, South Asia, and East Asia & Pacific region, hence interventions for the prevention and rehabilitation of GND should target children with CM in Sub-Saharan Africa. Most GND at hospital discharge are likely markers of acute illness and are transient in nature. A number of clinical factors associated with the pathophysiology of CM increase the risk of acute and chronic GND. This calls for close monitoring of children with CM by clinicians, during admission and targeting treatment protocols suited to each child.

## Supporting information

**S1 Protocol. Prevalence and risk factors of gross neurologic deficits in children after severe malaria: a systematic review protocol.**
(PDF)

**S1 Checklist. PRISMA 2020 checklist.**
(DOCX)

**S1 Table. List of excluded studies on full text.**
(DOCX)

**S1 Fig. ROBINS-E Traffic plot.**
(TIF)

**S2 Fig. ROBINS-E Summary plot.**
(TIF)

**S3 Fig. Funnel plot.**
(TIF)

**S4 Fig. Prevalence of gross neurologic deficits among children with Severe malaria at hospital discharge in studies with low and moderate risk of bias.**
(TIF)

**S5 Fig. Prevalence of gross neurologic deficits among children with Severe malaria at hospital discharge by risk of bias.**
(TIF)

## Acknowledgments

I acknowledge the Amsterdam University Medical Centers, University of Amsterdam, Global Health Uganda and the Africa Center for Systematic Reviews and Knowledge Translation for all the support and guidance in writing this manuscript. I acknowledge support from the University of Oxford in providing grey literature on the total number of children who had GNDs (motor deficits, vision deficits, hearing and speech deficits) at hospital discharge and at 28 days post-discharge, for one of the manuscripts (Dondorp, 2010) included in the meta-analysis.

## Author contributions

**Conceptualization:** Allen Eva Okullo, Chandy C. John, Michael Boele van Hensbroek, Richard Idro.

**Data curation:** Allen Eva Okullo, Simple Ouma, Kevin Ouma Ojiambo, Caroline Otike, Alison Annet Kinengyere.

**Formal analysis:** Allen Eva Okullo, Simple Ouma, Kevin Ouma Ojiambo.

**Methodology:** Allen Eva Okullo, Simple Ouma, Kevin Ouma Ojiambo, Caroline Otike, Andrea L. Conroy, Moses Ocan, Ekwaro A. Obuku.

**Project administration:** Allen Eva Okullo.

**Software:** Kevin Ouma Ojiambo, Alison Annet Kinengyere.

**Supervision:** Allen Eva Okullo, Chandy C. John, Michael Boele van Hensbroek, Richard Idro.

**Validation:** Allen Eva Okullo, Simple Ouma, Chandy C. John, Michael Boele van Hensbroek, Richard Idro.

**Visualization:** Kevin Ouma Ojiambo, Andrea L. Conroy, Moses Ocan, Ekwaro A. Obuku.

**Writing – original draft:** Allen Eva Okullo.

**Writing – review & editing:** Allen Eva Okullo, Simple Ouma, Chandy C. John, Michael Boele van Hensbroek, Kevin Ouma Ojiambo, Caroline Otike, Alison Annet Kinengyere, Andrea L. Conroy, Moses Ocan, Ekwaro A. Obuku, Richard Idro.

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
