## [Decision Letter · Decision Letter 0]

9 Jan 2026

Dear Dr. Okullo,

We look forward to receiving your revised manuscript.

Kind regards,

Alejandro Torrado Pacheco, PhD

Staff Editor

PLOS One

Journal Requirements:

**Additional Editor Comments:**

The manuscript has been evaluated by two reviewers, and their comments are available below.

Could you please carefully revise the manuscript to address all comments raised?

Reviewers' comments:

Reviewer's Responses to Questions

**Comments to the Author**

1. Is the manuscript technically sound, and do the data support the conclusions?

Reviewer #1: Yes

Reviewer #2: Yes

2. Has the statistical analysis been performed appropriately and rigorously?

Reviewer #1: Yes

Reviewer #2: Yes

3. Have the authors made all data underlying the findings in their manuscript fully available?

Reviewer #1: Yes

Reviewer #2: No

4. Is the manuscript presented in an intelligible fashion and written in standard English?

Reviewer #1: Yes

Reviewer #2: Yes

Reviewer #1: Line 255: Please report the time frame covered in the search.

Lines 273 - 274: The authors will do well to separate motor and movement deficits.

Table 1: Please citeLines the references properly e.g. Schmutzhard et al (1984).

Lines 331 - 337: It is important to re-state that these wereamong studies reporting frequencies for the respective periods which were mutually exclusive. Otherwise it would appear that frequency of CND was higher at after 6 months compared to after 12months and >12 months.

Lines 345 - 347: Revise to - compared to South Asia 6.1% (95%CI: 0.0-14.8) (I2= 0.0%) and East Asia & Pacific, 2.3% (95%CI: 0.8-3.8) (I2= 0.0%) and (Fig 7).

Lines 360 - 380: These 2 paragraphs should be better written for better clarity.

Reviewer #2: This paper reviews the literature on gross neurologic deficits in children and makes two primary contributions: (1) a meta-analysis of the prevalence of gross neurologic deficits (GND), and (2) a review of associated risk factors. Overall, the manuscript is clearly written, and the analyses are comprehensive and reasonably robust. The results appear to support the authors’ conclusions. I have a few minor suggestions.

As acknowledged by the authors, there is substantial heterogeneity in prevalence estimates. It may be worth considering adjustment for study-level characteristics (e.g., study design) in the random-effects model. Meta-analyses often account for factors such as whether studies were prospective, whether outcome assessment was blinded, or how cases were ascertained. If the authors choose not to include such fixed effects, they should briefly discuss and justify this decision.

It would be helpful to consider whether there is a temporal effect on prevalence estimates. For example, differences across studies may reflect the time periods during which they were conducted, particularly if diagnostic criteria or clinical practices evolved over time.

It was not entirely clear whether treatment protocols differed across studies. Variation in treatments could plausibly contribute to differences in observed GND prevalence, and a brief discussion of this issue—or clarification if treatment was relatively uniform—would strengthen the interpretation of the findings.

**Do you want your identity to be public for this peer review?** For information about this choice, including consent withdrawal, please see our Privacy Policy

Reviewer #1: **Yes:** Babatunde O. Ogunbosi

Reviewer #2: No

---

## [Author Response · Author response to Decision Letter 1]

19 Feb 2026

[PONE-D-25-49673] PLOS ONE

Responses to comments from the reviewers:

Reviewer #1:

1. Line 255: Please report the time frame covered in the search.

Response:

Thank you for this comment. We have now reported the time frame covered in the search as follows: ‘1946 upto 11 June 2023’, in line 255 of the manuscript.

2. Lines 273 - 274: The authors will do well to separate motor and movement deficits.

Response:

Thank you for this comment. We have separated motor and movement deficits in line 273-274. The lines now read as follows:

‘motor deficits such as paresis, plegia, hypertonia, hypotonia, dystonia, divergent squint, oral dyspraxia; movement deficits such as ataxia’.

3. Table 1: Please cite Lines the references properly e.g. Schmutzhard et al (1984).

Response:

Thank you. We have cited the references properly in both table 1 & 2 as guided.

4. Lines 331 - 337: It is important to re-state that these were among studies reporting frequencies for the respective periods which were mutually exclusive. Otherwise it would appear that frequency of CND was higher at after 6 months compared to after 12months and >12 months.

Response:

Thank you for this comment. We have revised lines 331-337 to show that the prevalence was derived from studies reporting GND in the respective periods which were mutually exclusive. In this regard, lines 334-335 in the manuscript now read as follows:

‘Prevalence of post-discharge GND among children with CM was 3.9% (95%CI: 2.5-5.2) among studies that reported GND at six months follow-up (I2= 47.0%) (Fig 4).’

While lines 338-340 now read as follows:

‘At 12 and over 12 months follow-up, prevalence of GND among children with CM was not statistically different at 5.1% (95%CI: 0.9-9.4) (I2= 47.5%) (Fig 5), and 5.7% (95%CI: 1.2-10.2) (I2= 79.2%) (Fig 6) respectively among studies that reported GND in these periods.’

5. Lines 345 - 347: Revise to - compared to South Asia 6.1% (95%CI: 0.0-14.8) (I2= 0.0%) and East Asia & Pacific, 2.3% (95%CI: 0.8-3.8) (I2= 0.0%) and (Fig 7).

Response:

Thank you. I have made the revision as guided, currently in lines 348-349 of manuscript.

6. Lines 360 - 380: These 2 paragraphs should be better written for better clarity.

Response:

Thank you for this insight. I have revised the three paragraphs from line 351-380 to provide a much clearer paragraph on the risk factors for gross neurologic deficits in children with cerebral malaria globally. The revised paragraph on lines 372-390 in the manuscript, reads as follows:

Nine studies reported adjusted odds ratios (AORs) for risk factors of GND following childhood cerebral malaria (3, 7, 11, 12, 15, 16, 17, 42, 48) (Table 2). Risk factors associated with GND at hospital discharge included: focal convulsion, AOR=11.3, 95%CI: 5.5-23.2 (48); unresolved acute kidney injury (AOR=3.7, 95%CI: 1.9-7.2) along with elevated blood urea nitrogen (BUN) (>20mg/dL) (AOR=2.8, 95%CI: 1.7-4.9) (16); metabolic acidosis, AOR=4.5, and elevated plasma creatinine, AOR=10.5 (7), suggestive of acute kidney injury, which were associated with short-term deficits at 1 month. Some risk factors were associated with GND both at hospital discharge, and long-term (GND persisting >3months duration of follow-up). These included: admission with profound/deep coma, AOR=5.5, 95%CI: 1.5-20.7 (42), AOR= 7.4, 95%CI: 1.8-29.7 (11) and AOR=28.8, 95%CI: 3.0-280 (3); coma lasting for �48hours, AOR=13.4, 95%CI: 3.4-52.4 (11) and AOR=3.8, 95%CI: 1.9-7.9 (48); multiple convulsions, AOR=7.1, 95%CI: 2.2-22.7 (11), AOR=12.8, 95%CI: 3.0–211 (12), and AOR=16.4, 95%CI: 2.9-90.8 (42); hypoglycemia, AOR=6.1, 95%CI: 1.4-25.7 (3) and AOR=3.2, 95%CI: 1.5-6.9 (15) and acute kidney injury, AOR=3.0, 95%CI: 1.2-7.6 (17) and AOR= 2.0, 95%CI:1.2-3.4 (16) as shown in Table 2. Risk factors associated with long-term GND included: previous admissions (AOR=4.8, 95%CI: 1.3-17.6), focal neurological signs observed during admission (AOR=12.3, 95%CI: 1.4-110), multiple seizures (AOR=8.3, 95%CI: 2.3-29.5), previous history of seizures (AOR=7.6, 95%CI: 1.7-33.6), age below 3years (AOR=6.4, 95%CI: 1.2-36), severe malnutrition (AOR=6.6, 95%CI: 1.4-30.6), and features of raised intracranial pressure (AOR=6.0, 95%CI: 1.4-30.6) (3).

Reviewer #2: This paper reviews the literature on gross neurologic deficits in children and makes two primary contributions: (1) a meta-analysis of the prevalence of gross neurologic deficits (GND), and (2) a review of associated risk factors. Overall, the manuscript is clearly written, and the analyses are comprehensive and reasonably robust. The results appear to support the authors’ conclusions. I have a few minor suggestions.

• As acknowledged by the authors, there is substantial heterogeneity in prevalence estimates. It may be worth considering adjustment for study-level characteristics (e.g., study design) in the random-effects model. Meta-analyses often account for factors such as whether studies were prospective, whether outcome assessment was blinded, or how cases were ascertained. If the authors choose not to include such fixed effects, they should briefly discuss and justify this decision.

Response:

Thank you. We have performed a sub-group analysis by study design and included a write-up on the results using the random-effects model on lines 352-360, page 21 of the manuscript, under the title ‘Prevalence of gross neurologic deficits among children with severe malaria by study design’. It was not possible to perform a sub-analysis by blinding status because studies did not include this information.

• It would be helpful to consider whether there is a temporal effect on prevalence estimates. For example, differences across studies may reflect the time periods during which they were conducted, particularly if diagnostic criteria or clinical practices evolved over time.

Response:

Thank you for this comment. A temporal effect on the prevalence due to changes in the diagnostic criteria could not be ascertained since the studies included did not provide differences in the criteria used to diagnose severe malaria and in particular cerebral malaria. Most studies that included information on the diagnostic criteria used the WHO standard case definition characterized in part by a positive blood slide by microscopy.

• It was not entirely clear whether treatment protocols differed across studies. Variation in treatments could plausibly contribute to differences in observed GND prevalence, and a brief discussion of this issue—or clarification if treatment was relatively uniform—would strengthen the interpretation of the findings.

Response:

Thank you for this insight.

Treatment protocols did indeed differ across studies. We have included a statement on the different types of treatments that were reported in the different studies on lines 281- 284, under characteristics of included studies on page 14 of the manuscript. The statement reads as follows: ‘Among studies that reported on treatment, types of treatment included: chloroquine (IV or subcutaneous) (n=2), chloroquine or IV quinine (n=2), IV quinine (n=6), ACTs (IV artesunate or IM Artemether) or IV quinine (n=8) and ACTs (n=4).’

We also performed a sub-group analysis by type of treatment using the random-effects model, however the difference in prevalence of GNDs by treatment type was not statistically significant (P=0.67) (I2=95%). We have included a write-up on the results on lines 361-368 on page 22, under the title ‘Prevalence of gross neurologic deficits among children with severe malaria by type of treatment’.

---

## [Decision Letter · Decision Letter 1]

2 Mar 2026

Prevalence and risk factors of gross neurologic deficits in children after severe malaria: A systematic review and meta-analysis

PONE-D-25-49673R1

Dear Dr. Okullo,

We’re pleased to inform you that your manuscript has been judged scientifically suitable for publication and will be formally accepted for publication once it meets all outstanding technical requirements.

Kind regards,

Bashir Sajo Mienda, PhD

Academic Editor

PLOS One

Additional Editor Comments (optional):

Reviewers' comments:

Reviewer's Responses to Questions

**Comments to the Author**

Reviewer #1: All comments have been addressed

Reviewer #2: All comments have been addressed

2. Is the manuscript technically sound, and do the data support the conclusions?

Reviewer #1: Yes

Reviewer #2: Yes

3. Has the statistical analysis been performed appropriately and rigorously?

Reviewer #1: Yes

Reviewer #2: Yes

4. Have the authors made all data underlying the findings in their manuscript fully available?

Reviewer #1: Yes

Reviewer #2: Yes

5. Is the manuscript presented in an intelligible fashion and written in standard English?

Reviewer #1: Yes

Reviewer #2: Yes

Reviewer #1: The comments raised in the earlier review have been addressed by the authors.

Congratulations on this important manuscript.

Thank you for attending to the comments raised.

Reviewer #2: Authors have addressed my concerns. The additional analyses are fine and appropriate. I recommend publication.

**Do you want your identity to be public for this peer review?** For information about this choice, including consent withdrawal, please see our Privacy Policy

Reviewer #1: **Yes:** Babatunde Ogunbosi

Reviewer #2: No

---

## [Editor Report · Acceptance letter]

PONE-D-25-49673R1

PLOS One

Dear Dr. Okullo,

I'm pleased to inform you that your manuscript has been deemed suitable for publication in PLOS One. Congratulations! Your manuscript is now being handed over to our production team.

Kind regards,

on behalf of

Dr. Bashir Sajo Mienda

Academic Editor

PLOS One